# Comparison between the Effects of Adding Vitamins, Trace Elements, and Nanoparticles to SHOTOR Extender on the Cryopreservation of Dromedary Camel Epididymal Spermatozoa

**DOI:** 10.3390/ani10010078

**Published:** 2020-01-02

**Authors:** Mohamed A. Shahin, Wael A. Khalil, Islam M. Saadeldin, Ayman Abdel-Aziz Swelum, Mostafa A. El-Harairy

**Affiliations:** 1Department of Animal Production, Faculty of Agriculture, Mansoura University, Mansoura 35516, Egypt; 2Electron Microscope Unit, Mansoura University, Mansoura 35516, Egypt; 3Department of Animal Production, College of Food and Agricultural Sciences, King Saud University, Riyadh 11451, Saudi Arabia; aswelum@ksu.edu.sa; 4Department of Physiology, Faculty of Veterinary Medicine, Zagazig University, Zagazig 44519, Egypt; 5Department of Theriogenology, Faculty of Veterinary Medicine, Zagazig University, Zagazig 44519, Egypt

**Keywords:** semen, freezing, nanoparticles, vitamins, camel

## Abstract

**Simple Summary:**

This is a comprehensive study to compare between the effects of different supplements (vitamins C and E, trace elements Na_2_SeO_3_ and ZnSO_4_, and nanoparticles of zinc oxide and selenium) to the semen extender of camel epididymal spermatozoa during cooling and freezing/thawing cryopreservation. Supplementation of the semen SHOTOR extender with zinc oxide and selenium nanoparticles lead to improved progressive motility, vitality, and anti-oxidative defense, and reduced the ultrastructural abnormalities in camel epididymal spermatozoa.

**Abstract:**

There are several obstacles in camel semen cryopreservation; such as increasing semen viscosity and the reduction in motile spermatozoa after ejaculation. Epididymal spermatozoa offer an efficient alternative to overcome these problems and are well-suited for artificial insemination in camels. In the current study, we compared the effects of supplementation with vitamin C, E, inorganic trace elements of selenium (Na_2_SeO_3_) and zinc (ZnSO_4_), and zinc and selenium nanoparticles (ZnONPs and SeNPs, respectively) on the cryopreservation of dromedary camel epididymal spermatozoa. When the SHOTOR extender was supplemented with ZnONPs and SeNPs; the sperm showed increased progressive motility; vitality; and membrane integrity after cooling at 5 °C for 2 h; when compared to the control and vitamin-supplemented groups. Moreover, the ZnONPs and SeNPs supplementation improved the progressive motility, vitality, sperm membrane integrity, ultrastructural morphology, and decreased apoptosis when frozen and thawed. SeNPs significantly increased reduced glutathione (GSH), superoxide dismutase (SOD), and decreased lipid peroxide malondialdehyde (MDA) levels. The advantageous effects of the trace elements were potentiated by reduction into a nano-sized particle, which could increase bioavailability and reduce the undesired liberation of toxic concentrations. We recommend the inclusion of SeNPs or ZnONPs to SHOTOR extenders to improve the cryotolerance of camel epididymal spermatozoa.

## 1. Introduction

The camel (*Camelus dromedarius*) is one of the oldest known mammals to have adapted to desert climates and is used for milk and meat production [1], as well as a sports animal [2]. Compared to other farm animals, studies on camel reproductive physiology are lacking because of the many complex genetic and environmental factors that contribute to the declined fertility in camels. In recent years, the number of studies on the camelid family, in terms of science and research, has greatly increased [3]. There are several ways to improve the productivity and reproductive performance of the Arabian camel [4,5,6,7,8], such as artificial insemination (AI). AI is a useful tool for genetic livestock improvement but has not yet been optimized in camels, owing to the improper protocols for camel semen cryopreservation [9,10]. The obstacles surrounding this protocol include (1) difficulties in collecting semen from aggressive males in rut, (2) the absence of suitable extenders for its storage, (3) low sperm concentration, (4) low sperm motility, and (5) the viscous nature of semen [1,11,12].

Epididymal spermatozoa, on the other hand, could be used as an alternative method for camel AI [11]. Caudal epididymal sperm collection is an important technique in the generation and conservation of animal specimens and is particularly useful when the animal is seriously injured or when collecting from dead specimens [13].

The cryopreservation of camel epididymal spermatozoa involves a suitable cryo-diluent to protect the sperm from cryo-damage [14]. To date, no studies have systematically examined the ability of the epididymis to protect sperm from oxidative stress (e.g., by epidermal fluids containing antioxidants) [11].

The mammalian sperm cell contains a high ratio of polyunsaturated fatty acids, and is therefore highly susceptible to peroxidative damage and oxidative stress (OS), particularly after cryopreservation. The cryopreservation process may result in a loss of membrane integrity, DNA fragmentation, or impaired cell function, all of which may cause decreased sperm movement and reduced fertilization capacity [15,16,17]. OS is a major factor affecting male fertility and results from an imbalance between the production of antioxidant defense mechanisms and reactive oxygen species (ROS) in the cell.

In recent years, studies have also been conducted on camels, cow bulls, and human semen diluents, to supplement anti-oxidants such as glutathione, vitamin C (Vit C), vitamin E (Vit E), zinc (Zn), and selenium (Se) to improve the post-thawed motility, vitality, and membrane integrity of spermatozoa [18,19,20].

Vit C is a highly effective antioxidant and a free radical scavenger in many metabolic processes [21,22]. Vit C increases the percentage of live, acrosome-intact sperm, and decreases the percentage of abnormal sperm during storage at 5 °C. Vit C prevents membrane lipid oxidation during preparation and thus has protective effects. Chinoy et al. [23] reported that the testes and seminal plasma are extremely sensitive to decreased Vit C levels in the body.

Vit E, a lipophilic molecule in the cell membrane, is considered to be a membrane-stabilizer and a strong antioxidant agent that protects the cell membrane from lipid peroxidation and ROS attacks [24]. Vitamin E is the most widely known antioxidant and provides protective effects by reducing or preventing peroxidative damage [25]. Previous research has shown that Vit E can improve the post-thaw sperm quality in bulls [26], rabbits [27], roosters [28], and sheep [29].

Zn has antioxidant properties and reduces ROS released by defective spermatozoa and leucocytes, inhibits lipid peroxidation and reduces circulating anti-sperm antibodies [30,31]. The addition of 100 μM zinc sulfate to semen extender showed a significant increase in the percentage of intact DNA sperm, mitochondrial function, and progressive motility compared to cryopreserved semen samples without zinc supplementation [19]. Dorostkar et al. [32] reported that the addition of 0.288 mg/L zinc sulfate to the extender provides better sperm preservation during freezing processes compared to the control group, which in turn can lead to higher fertility in semen. Ghallab et al. [33] reported that adding 200 μM Zn to semen diluent may improve the quality of frozen/thawed and chilled Arab stallion spermatozoa parameter.

The antioxidant function of Se is mediated through glutathione peroxidase enzyme activity that is known as an important antioxidant and a marker of oxidative stress and protects germ cells, proteins and organic membranes from OS [34]. Extenders containing 1 and 2 μg/mL of sodium selenite significantly improved buffalo frozen/thawed sperm motility, viability, membrane integrity, and total antioxidant capacity and reduced sperm DNA damage [32]. Marai et al. [35] reported that 0.1 ppm sodium selenite improved semen performance in rams by increasing semen volume per ejaculate, sperm motility, and concentration, and by decreasing the percentage of dead sperm, and sperm and acrosome damage abnormalities.

Nanoparticles (NPs), defined by having at least one dimension within the range of 1–100 nm, have become increasingly common in a variety of medical areas [20]. However, there are concerns regarding their biological impact. Recently, nano-elements have drawn great interest due to their low toxicity and high bioavailability. This is because the nanometer particulates exhibit novel characteristics, such as a specific surface area, numerous active surface centers, high surface activity, high catalytic efficiency, and strong adsorption ability [36,37].

Zinc oxide nanoparticles (ZnONPs) increases antioxidant enzymes and improves the quality of sperm [38], and is therefore considered an essential element for spermatogenesis. Zinc (Zn) deficiency in rats triggers a decline in ribonucleic acid (RNA), deoxyribonucleic acid (DNA), and protein levels, and is accompanied by increased ribonuclease activity. Zn functions as a cofactor for DNA and RNA polymerase operations and RNA-dependent DNA polymerase and is therefore important for cell growth [39,40]. Zinc is involved in sperm motility, through adenosine triphosphate processes and phospholipid control, and is highly concentrated in the mature sperm tail [39]. Zinc plays an important role in sperm vitality and motility, affecting protein metabolism, nucleic acid synthesis, and the stabilization of the sperm membrane [41]. Zn can also neutralize the effects of ROS, thereby increasing the efficiency of ATP pathways [42]. Zn has antioxidant properties (it reduces the ROS released by defective spermatozoa and leucocytes), inhibits lipid peroxidation (by phospholipase inhibition), and reduces the amount of circulating anti-sperm antibodies [43,44].

Selenium (Se) is also an important trace element that is fundamental to human health. It plays a role in cell development, apoptosis, and cell signaling mechanisms, and several studies have demonstrated its protective antioxidant characteristics [45]. Se is a powerful antioxidant that alters the expression of selenoproteins-it is incorporated by replacing sulfur in proteins with seleno-amino acids (L-selenomethionine, L-selenocysteine) and selenoenzymes, such as GPxs. Glutathione peroxidases 4 (GPXs4) is a critical component affecting sperm quality and male fertility. Therefore, spermatozoa may be vulnerable to oxidative stress if the Se content of the selenoproteins is low [46]. Several studies have used nano-selenium (SeNPs) as a ROS scavenger to safeguard against oxidative damage in sperm cells. Adding SeNPs to the semen extender enhanced the post-thawing quality and oxidizing rooster semen variables [28,47]. The oral supplementation of SeNPs also protected the spermatozoa quality (motility, DNA integrity) and spermatogenesis against oxidative damage caused by cisplatin, a male reproductive toxicant [48].

Although prior studies have compared various camel semen extenders and measured sperm motility, morphology, and vitality [49,50,51], few have used electron microscopy to examine frozen-thawed camel spermatozoa [52,53,54]. Therefore, the current study aimed to evaluate the freezability and molecular functional integrity of dromedary camel epididymal spermatozoa after dilution by extender supplemented with vit C, vit E, inorganic trace elements (Na_2_SeO_3_ and ZnSO_4_), and nano-sized zinc or selenium.

## 2. Materials and Methods

### 2.1. Characteristics of Nano-Sized Elements (Particle Size, Zeta Potential, and Ultramorphology)

SeNPs were purchased from Nanocs (cat. no. Se50-01-5, Nanocs, New York, NY, USA) and Zinc oxide nanoparticles (ZnONPs) were purchased from Sigma-Aldrich (cat. no. 544906, Sigma-Aldrich, Taufkirchen, Germany). Photon correlation spectroscopy Malvern zetasizer Nano-Zs90 was used to measure the particle size and polydispersity index (PDI) of the Zn and Se NPs. The samples were suitably diluted and sonicated to uniformly distribute the particles. All measurements were done in triplicate, and the results were calculated as the mean ± standard deviation (SD). The zeta potential of the Zn and Se NPs was measured by photon correlation spectroscopy at 25 °C. To determine the NPs’ surface charge, the samples were properly diluted with double deionized water, placed in an electrophoretic cell, and measured. The morphological evaluation of Zn and Se NPs was observed by transmission electron microscopy (TEM) (JEOL-JEM-2100) at 160 kV (EM-Unit at Mansoura University, Mansoura, Egypt). For this procedure, one milliliter of the NP sample dispersion was properly diluted with double deionized water and sonicated for 2 min using an ultrasonic bath. After dilution, one drop of the NPs was added to a carbon-coated copper grid, and the excess material was removed, leaving a thin film stretched over the holes. This was allowed to dry at room temperature for 10 min before the image was captured and analyzed in the Gatan software (Version 2.11. 1404.0, Pleasanton, CA, USA).

The average diameter of the ZnONPs and SeNPs was 30.92 ± 1.25 nm and 78.47 ± 17.93 nm, respectively, with low PDI values, suggesting a narrow size distribution (Appendix A). The ZnONPs had a positive zeta potential (32.16 ± 0.252 mV), while the SeNPs had a negative zeta potential (−20.36 ± 1.79 mV). The representative size and zeta potential curves of the ZnONPs and SeNPs are depicted in Appendix A, respectively. Moreover, the TEM images exhibited a spherical nanoscopic size for the ZnONPs and SeNPs, in agreement with the Zetasizer measurements (Figure 1).

### 2.2. Epididymal Camel Spermatozoa Collection

One hundred and two testes from healthy, mature dromedary camels, 6 to 12 years old, were collected from a local abattoir during the breeding season (January–May 2019). There was no member of the research team involved in pre-slaughter live animal handling or in the process of slaughtering, hence ethical approval is not required. The testes were immediately placed in a plastic bag containing normal saline and kept in an air-tight sterile cryo-box at 5 °C. All samples were processed within 6 h after collection. In the laboratory, the testes were washed with sterile warm saline, and the epididymis was isolated. The epididymis was washed three times with warm saline and once with 70% ethyl alcohol. Various incisions in the corpus and tail of epididymis were performed with a scalpel, and the spermatozoa were released by manually pressing the dissected epididymis. The sperm was collected by rinsing 3–4 times with a sterile disposable syringe containing 5 mL warm extender. The recovered spermatozoa were placed in 50 mL tubes.

### 2.3. Experimental Design

The Tris-based extender, SHOTOR^®^ diluent [51], comprised of 2.6 g Tris, 1.35 g citric acid, 1.2 g glucose, 0.9 g fructose, 1000 IU/mL penicillin, and 1000 µg/mL streptomycin dissolved in 100 mL of deionized water. The osmolality was 330 mOsm/kg, and the pH was 6.9. The extender was composed of 74 mL buffer + 6 mL glycerol + 20 mL egg yolk. The following supplements were added to the SHOTER extender to examine the effects on spermatozoa cryopreservation: Vit E (α tocopherol, T3251) at 200 μM/mL [55], Vit C (ascorbic acid, A4544) at 1 mg/mL [18], SeNPs at 1 μg/mL [47], ZnONPs at 50 μg/mL [20], Na_2_SeO_3_ (Loba Chemie Pvt. Ltd., Mumbai, India) at 2 μg/mL [32], and ZnSO_4_ (Loba Chemie) at 100 μM/mL [19].

Fluid rich spermatozoa collected from the cauda and corpus epididymides were initially evaluated for progressive motility and sperm cell concentration. Fluid rich spermatozoa having ≥ 60% motility was diluted with different extenders (final concentration: 80 × 10^6^ sperm/mL), and the sperm characteristics were evaluated, including progressive motility, vitality, abnormal morphology, and the hypo-osmotic swollen test. The diluted epididymal spermatozoa were cooled to 5 °C for a period of 2 h (for equilibration) before being loaded into 0.25 mL straws. The straw was placed 4 cm above liquid nitrogen vapor for 10 min and then immersed in liquid nitrogen. The straws remained in liquid nitrogen until thawing at 37 °C in a water bath for 30 sec.

### 2.4. Epididymal Spermatozoa Evaluation

#### 2.4.1. Sperm Progressive Motility

Equilibrated and post-thawed epididymal spermatozoa were evaluated for progressive motility rate under a phase-contrast microscope (Leica DM 500) supplied with a hot stage adjusted to 37 °C [56].

#### 2.4.2. Sperm Vitality and Abnormalities

A smear of the diluted semen samples was placed on a glass slide and stained with a dual staining procedure; eosin (5%) and nigrosin (10%) [57]. A total of 200 spermatozoa from each sample were examined with a light microscope at 400× magnification (Leica DM 500, Leica Mikrosysteme Vertrieb GmbH, Wetzlar, Germany). For each sample, the number of dead spermatozoa (red-stained) was counted, and the morphological abnormalities of the spermatozoa were determined, i.e., spermatozoa bearing head, tail, and cytoplasmic droplets abnormalities [58].

#### 2.4.3. Sperm Plasma Membrane Integrity

The hypo-osmotic swelling test (HOST) was used to evaluate the functional plasma membrane of spermatozoa [59]. Briefly, 10 µL of semen was incubated with 100 µL hypo-osmotic solution (6.75 g/L fructose and 3.67 g/L sodium citrate, for an osmolality of 75 mOsmol/L) at 37 °C for 30 min. Afterward, 10 µL of the mixture was placed on a microscope slide and mounted with a coverslip. A total of 300 spermatozoa (from each sample) were evaluated for swollen and coiled tails under phase-contrast microscopy (Leica DM 500, Leica Mikrosysteme Vertrieb GmbH) at 400× magnification.

#### 2.4.4. Antioxidants Assay

The post-thawed epididymal spermatozoa samples were centrifuged for 15 min at 1500 rpm, and the extender was separated and stored at −20 °C. The concentrations of reduced glutathione (GSH) [60,61], malondialdehyde (MDA), and superoxide dismutase (SOD) [62] were analyzed by commercial kits (Biodiagnostic, Cairo, Egypt) using a spectrophotometer (Spectro UV-VIS Auto, UV-2602, Labomed, Los Angeles, CA, USA).

#### 2.4.5. Assessment of Sperm Apoptosis and Necrosis through Flowcytometric Analysis

The epididymal spermatozoa samples were processed for annexin V staining as described in Chaveiro et al. [63], but with some modifications. Briefly, 1 mL of sperm suspension was added to a 5 mL tube and suspended in a 2 mL binding buffer and thoroughly mixed. 100 µL of the sperm suspension was mixed with 5 µL of annexin V (FITC label) and then 5 µL propidium iodide (PI, PE label), and incubated for at least 15 min, in darkness at room temperature. After incubation, the sperm was suspended in 200 µL binding buffer. The flowcytometric analysis was performed on an Accuri C6 Cytometer (BD Biosciences, San Jose, CA, USA) using the Accuri C6 software (Becton Dickinson) for data acquisition and analysis [64]. The percentages of annexin V negative or positive (A− or A+), PI negative or positive (PI− or PI+), and double-positive cells were evaluated. As described by Peña et al. [65], four different categories of spermatozoa were determined: (1) viable (A−/PI−), with no fluorescence signal, and recorded as live without membrane dysfunction (live sperm); (2) early apoptotic (A+/PI−), but viable spermatozoa, labeled with annexin V but not with PI (live sperm); (3) late apoptotic spermatozoa (A+/PI+) labeled with annexin V and PI and with damaged permeable membranes (dead sperm); (4) necrotic spermatozoa (A−/PI+), labeled by PI but not annexin V, that have completely lost the sperm membrane (dead sperm).

#### 2.4.6. Assessment of Sperm Morphology Using Scanning Electron Microscope (SEM)

The specimens were centrifuged at 500× *g* for 20 min, and the sperm pellets were collected. The samples were fixed in a solution containing 2.5% (w/v) buffered glutaraldehyde and 2% (w/v) paraformaldehyde in a 0.1 M sodium phosphate buffer (pH 7.4) at 4 °C overnight [66]. The specimens were then washed three times for 15 min each in 0.1 M sodium phosphate and treated with a 2% sodium phosphate-buffered osmium tetroxide (pH 7.4) for 90 min. Finally, the specimens were washed with a 0.1 M sodium phosphate buffer (pH 7.4) and dehydrated in an increasing gradient of ethanol. Four drops of 100% acetone were added to the specimen on small glass plates glued to the specimen stubs of the microscope. After the acetone had evaporated, the specimens were coated with gold-palladium membranes and observed in a Jeol-JSM-6510 L.V SEM. The microscope was operated at 20 kV, and only the central areas of the glass plates were examined.

#### 2.4.7. Assessment of Sperm Ultrastructure Using Transmission Electron Microscope (TEM)

The samples were processed for transmission electron microscopy (TEM) according to Heath et al. [53]. Briefly, the straws from each treatment were washed three times by centrifugation at 1000 rpm for 5 min using Phosphate Buffered Saline, and suspended in a fixative solution of 2.5% (w/v) buffered glutaraldehyde and 2% (w/v) paraformaldehyde in a 0.1 M sodium phosphate buffer (pH 7.4) for 2 h at 4 °C. The samples were then washed and post-fixed in 1% osmium tetroxide for 1 h at room temperature in darkness. The fixed samples were dehydrated in an ethanol gradient, treated with acetone, embedded in an Epon resin (Epon 812; FlukaChemie, Switzerland), and ultrathin-sectioned (60–80 nm) for TEM. Ultrathin sections were observed using a JEOL 2100 TEM at 80 kV. The sperm ultrastructure (acrosome, plasma membrane, and mid-piece) was examined in 100 sperm cells per treatment. 

### 2.5. Statistical Analysis

All data were statistically analyzed by one-way ANOVA design using a software package (SAS, 2007, Cary, NC, USA) [67]. Completely randomized design was used based on the following model: Yij = μ + Gi + eij Where μ = the overall mean, Gi = Treatment (1,2,...7), and eij = residual error. The percentages of values were transformed by arcsine values before analysis. Differences between groups were tested by Duncan’s multiple range test [68] and set at *p* < 0.05.

## 3. Results

### 3.1. Effects on Sperm Quality After Cooling (5 °C for 2 h) and Pre-Freezing

Compared to the control group, supplementation of the extender with nano-sized selenium and zinc resulted in significant increases in sperm progressive motility, vitality, and sperm membrane integrity after cooling at 5 °C for 2 h (Table 1). Supplementation with vitamins C and E did not affect vitality but did show advantageous effects on sperm progressive motility and membrane integrity (Table 1). There were no differences among the experimental groups for sperm abnormality and cytoplasmic droplets.

### 3.2. Effects on Post-Thawing Sperm Quality

Nano-sized selenium and zinc supplementation significantly improved the sperm properties post-thawing. Sperm progressive motility, vitality, and membrane integrity increased significantly in the nano-sized selenium and zinc treatments when compared to the regular salts, vitamins, and control groups (Table 2). Conversely, the abnormality occurrence was significantly decreased by the supplementation of nano-sized and regular trace elements salts, when compared to the vitamin-supplemented and control groups (Table 2).

### 3.3. Effects on Sperm Apoptosis and Necrosis (Annexin V/PI Assay) Post-Thawing

Annexin V- and PI-negative sperm were significantly increased in the nano-selenium group, compared to the nano-zinc, vitamin C, vitamin E, sodium selenite, zinc sulfate, and control groups (Table 3). Early apoptotic and apoptotic sperm was significantly increased in the control group when compared to the other groups; the lowest value was observed in the nano-selenium supplemented group (Table 3). Representative pictures of the flow cytometry analysis are shown in Appendix A.

### 3.4. Effects on Oxidative Stress of the Extender Post-Thawing

The levels of GSH and the activity of SOD were significantly increased in the nano-selenium and nano-zinc groups when compared to the other experimental and control groups. The lowest values were observed in the control group (Table 4). However, the level of MDA was significantly increased in the control group when compared to the other groups (Table 4).

### 3.5. Effects on Sperm Ultra-Morphological Characters of Plasma Membrane (PM) and Acrosome Post-Thawing

Figure 2 and Figure 3 show the normal and abnormal ultrastructure of the epididymal camel spermatozoa. There was a significant increase in the percentage of intact plasma membranes in the nano-selenium group, compared to the control and zinc sulfate-supplemented groups (Table 5). The occurrence of abnormal ultra-structures (swollen and slightly swollen plasma membrane) was variable among the experimental groups, but there was a significant increase in the swollen membrane of the control group (Table 5). The lost PM (Table 5) and acrosome ultra-morphology (Table 6) and mid-piece structures (data are not shown) showed no differences among the experimental groups.

## 4. Discussion

Because of the high level of unsaturated fatty acids in the sperm plasma membrane and inefficient free radical scavenging system because of scanty cytoplasm in the sperm, it is highly vulnerable to the oxidative damage [69]. Sperm cryopreservation and thawing increase ROS formation [70]. ROS increase, together with reduced antioxidants, can lead to oxidative stress (OS) [71], which negatively affects semen quality, induces detrimental effects on spermatozoa [72], which can result in extensive changes, lipid peroxidation, in the plasma membrane concerning their organization as well as the function [73,74,75]. Therefore, OS causes sperm DNA damage and reduces sperm motility, functional integrity, endogenous antioxidant enzyme activity, and fertility [76].

Our results revealed that adding Vit C, Vit E, Na_2_SeO_3_, ZnSO_4_, ZnONPs, or SeNPs can improve the post-thaw quality of the dromedary camel epididymal spermatozoa. This improvement can be explained by their ant-oxidative properties. Therefore, the highest MDA level and the lowest GSH and SOD levels were observed in the control group. This means that these additives can protect sperm from oxidative stress during cryopreservation which reflected on semen quality parameters including progressive motility, vitality, plasma membrane integrity, abnormality, and/or intactness of acrosome and plasma membrane. Oxidative stress induced at the membrane level is correlated to MDA level, which is the commonly used biomarker for membrane lipid peroxidation of omega-3 and omega-6 fatty acids level [72]. The antioxidant system consisting of catalase (CAT), glutathione peroxidase (GSH-PX), and superoxide dismutase (SOD) acts synergistically to increase GSH and reduce MDA, to preserve sperm motility and vitality against OS [15,77].

Nanotechnology, such as nano-zinc and nano-selenium, can be used to achieve bioactive properties in the reproduction, digestion, development, and freezing cells of various elements [78]. Recent studies have shown the beneficial effects of nanoparticles for sperm freezing in humans and animals by lowering the sperm chromatin damage and MDA levels [79,80,81]. However, no studies have examined the impact of zinc oxide or selenium nanoparticles on epididymal camel sperm. The significantly lowest level of MDA and highest level of GSH were observed in the ZnONPs and SeNPs groups in the present study that suggests that the nanoparticles efficiently scavenge the free radicals generated during the freezing/thawing process. Therefore, ZnONPs and SeNPs improved the progressive motility, vitality, and spermatozoa membrane integrity, and decreased apoptosis in frozen-thawed sperm. Nano-sized manufacturing potentiates the effects of these metals when compared to the regular size particles of zinc sulfate, and sodium selenite, respectively. The effect of the NPs is attributable to their small size (surface area and size distribution), chemical composition (purity, crystallinity, and electronic properties), surface structure (surface reactivity, surface groups, inorganic or organic coatings), solubility, shape, and aggregation. These factors give synthetic nanoparticles physicochemical features and a higher surface reactivity than their counterparts of the regular trace elements and vitamins [82,83]. The nanoparticles form ZnONPs thought to be had a more efficient interaction with protamines [79]. According to the TEM results of Isaac et al. [79], the nanoparticles (ZnONPs) were accumulated around the spermatozoa and did not internalize into the sperm. A similar observation was recorded after using the fluorescent ZnONPs; the viable sperm membrane is not permeable to ZnONPs; while, the fluorescence ZnONPs was observed only in the dead spermatozoa. The spermatozoa membrane has a net negative charge and the ZnONPs have a neutral charge. The ZnONPs cannot dissociate to ions which is probably the reason why they cannot enter into the sperm chromatin [79].

Our findings agree with an earlier report by Isaac et al. [79] who reported that ZnONPs have membrane protective function. ZnONPs supplementation leads to an increase in antioxidant enzymes and improves the quality of sperm [20,38,84,85]. The free radical scavenging function of zinc is well established through earlier in vivo and in vitro studies [38,86]. Moreover, it enhanced the post-thawing quality when supplemented to rooster semen extender [28,47] as well as protected the spermatozoa quality (motility, DNA integrity) and spermatogenesis against oxidative damage when administered orally [48]. Therefore, it is probable that the protection of membrane peroxidation by ZnONPs and SeNPs through the reduction of MDA and increasing the SOD and GSH help in mitigating the subsequent damaging effect on the macromolecules like DNA and crucial organelles like mitochondria [38,87,88]. Similar results were recorded by Isaac et al. [79] who concluded that incubation of human ejaculate with ZnONPs improved the quality of cryopreserved human semen, particularly, with respect to chromatin integrity without any adverse effect on their functional competence. The protective effect on sperm chromatin during freezing/thawing process seems to be due to the formation of a protective layer of ZnONPs around the spermatozoa, which can prevent lipid peroxidation at the membrane level [79].

Safa et al. [28] showed that the addition of 5 mg/mL of vitamin E to freezing diluents combined with 1% SeNPs improved the post-thawing sperm motility, vitality, and oxidative variables in rooster semen. Khalil et al. [47] recommended an increase in the supplementation concentration of SeNPs from 0.5 to 1.0 mg/mL to Tris-egg yolk extenders after observing an increase in the percentage of viable sperm, and a decrease in early apoptotic, apoptotic, and necrotic sperm in post-thawed bull semen. However, increasing the concentration of SeNPs to 1.5 mg/mL adversely affected the viable sperm and increased the amount of apoptotic and necrotic sperm.

Our results also revealed that sodium selenite group had a higher level of GSH and SOD and lower level of MDA than zinc sulfate group, which reflect on semen quality. This can be explained by the smaller sizes of sodium selenite particles compared with zinc sulfate in addition to different roles of Zn and Se in the sperm.

## 5. Conclusions

In spite of supplementation of SHOTOR extender with Vit C, Vit E, Na_2_SeO_3_, ZnSO_4_, ZnONPs or SeNPs can improve the post-thaw quality of dromedary camel epididymal spermatozoa, the selenium and zinc nanoparticles provide the best post-thaw quality. The results presented here highlight favorable sperm properties such as increased motility, vitality, and ultrastructure morphology after cooling or freezing-thawing. We recommend the inclusion of nano-selenium and nano-zinc to extenders to improve the cryotolerance of camel epididymal spermatozoa.

## Figures and Tables

**Figure 1 animals-10-00078-f001:**
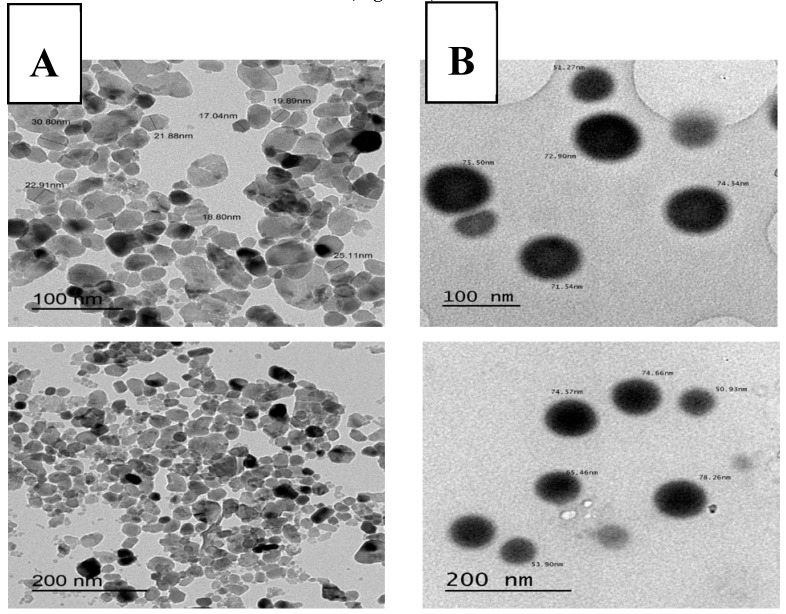
Transmission electron microscopy of (**A**) ZnONPs and (**B**) SeNPs with different magnifications.

**Figure 2 animals-10-00078-f002:**
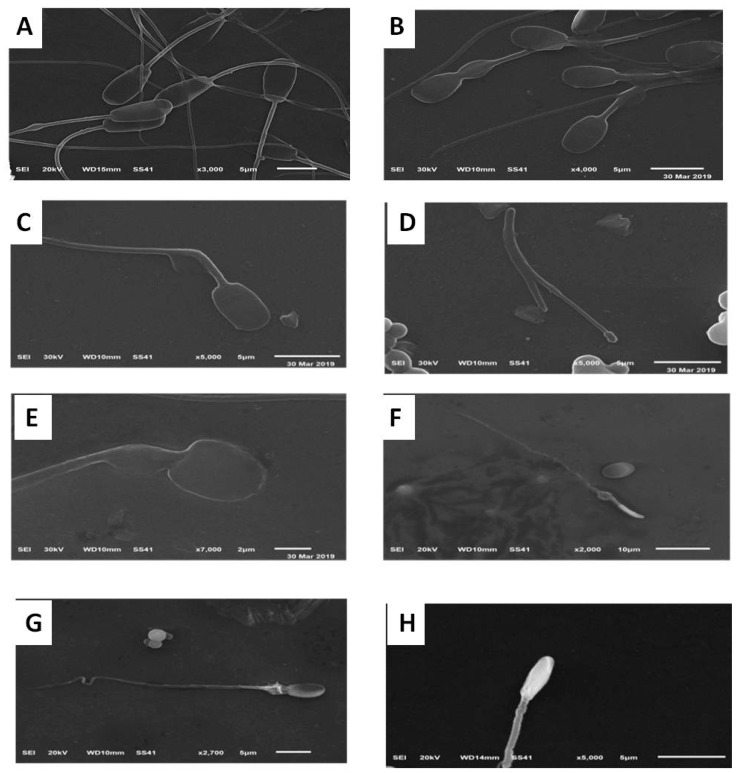
Representative scanning electron micrographs (SEM) showing the epididymal sperm morphology. Images (**A**,**B**) depict normal sperm cells while images (**C**–**H**) show abnormalities in the head and mid-piece (**C**). detached acrosomes; (**D**). detached head; (**E**), damaged plasma membrane and swelled head; (**F**). thin mid-piece and pen head; (**G**). small head; (**H**). abnormal head and mid-piece size).

**Figure 3 animals-10-00078-f003:**
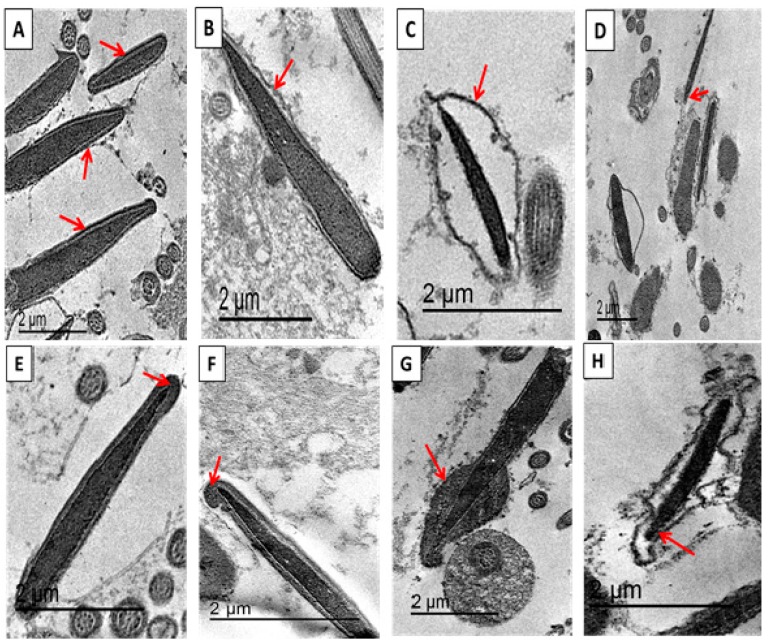
Transmission electron micrographs (80 kV) of longitudinal and ultrathin cross-sections of different epididymal camel spermatozoa after thawing. (**A**) Sperm with intact plasma membrane, (**B**) Sperm with slightly swollen plasma membrane, (**C**) Sperm with swollen plasma membrane, (**D**) Sperm with lost plasma membrane, (**E**) Sperm with intact acrosome, (**F**) Sperm with typical acrosome reaction, (**G**) Sperm with atypical acrosome reaction and (**H**) Sperm with lost acrosome.

**Table 1 animals-10-00078-t001:** Effect of supplementing SHOTOR extender with vitamins, trace elements, and nanoparticles on epididymal camel spermatozoa characteristics (%) after an equilibration period (2 h at 5 °C).

Treatment	Progressive Motility	Vitality	Plasma Membrane Integrity	Abnormality	Cytoplasmic Droplet
Control	60.8 ± 2.01 ^c^	63.2 ± 2.91 ^b^	60.2 ± 2.04 ^c^	12.3 ± 1.87	36.2 ± 1.51
Vitamin C	67.5 ± 2.14 ^ab^	70.2 ± 3.39 ^ab^	68.5 ± 1.61 ^ab^	11.0 ± 1.29	36.7 ± 1.69
Vitamin E	63.3 ± 1.67 ^bc^	65.5 ± 1.45 ^ab^	65.8 ± 0.31 ^b^	14.2 ± 1.68	34.5 ± 0.62
SeNPs	72.5 ± 1.44 ^a^	73.3 ± 0.85 ^a^	73.0 ± 1.83 ^a^	15.5 ± 1.50	35.3 ± 1.11
Na_2_SeO_3_	68.0 ± 1.22 ^ab^	69.8 ± 1.77 ^ab^	69.2 ± 2.48 ^ab^	11.6 ± 1.69	33.8 ± 1.71
ZnONPs	70.8 ± 2.01 ^a^	73.3 ± 2.14 ^a^	71.3 ± 1.87 ^ab^	12.5 ± 0.92	34.5 ± 1.48
ZnSO_4_	68.8 ± 2.39 ^ab^	72.0 ± 2.35 ^a^	70.5 ± 2.84 ^ab^	15.8 ± 2.53	33.8 ± 0.75

^a–c^ Means denoted within the same column with different superscripts are significantly different at *p* < 0.05.

**Table 2 animals-10-00078-t002:** Effect of supplementing SHOTOR extender with vitamins, minerals, and nanoparticles on sperm characteristics (%) in post-thawed epididymal camel spermatozoa.

Treatment	Progressive Motility	Vitality	Plasma Membrane Integrity	Abnormality	Cytoplasmic Droplet
Control	27.0 ± 1.22 ^c^	28.8 ± 1.32 ^c^	28.4 ± 1.12 ^c^	26.2 ± 1.02 ^a^	21.6 ± 2.84
Vitamin C	30.0 ± 1.58 ^c^	32.2 ± 2.11 ^c^	30.4 ± 2.36 ^c^	18.6 ± 0.75 ^b^	21.0 ± 2.61
Vitamin E	31.0 ± 1.00 ^c^	33.8 ± 0.86 ^c^	31.0 ± 0.89 ^c^	24.8 ± 0.86 ^a^	21.2 ± 2.71
SeNPs	48.3 ± 1.67 ^a^	50.7 ± 2.33 ^a^	50.0 ± 1.15 ^a^	18.7 ± 0.88 ^b^	20.7 ± 2.40
Na_2_SeO_3_	40.0 ± 2.04 ^b^	42.3 ± 2.14 ^b^	41.5 ± 2.87 ^b^	19.5 ± 0.65 ^b^	21.3 ± 2.46
ZnONPs	49.0 ± 1.87 ^a^	52.2 ± 2.15 ^a^	51.0 ± 2.19 ^a^	18.8 ± 0.58 ^b^	19.4 ± 1.81
ZnSO_4_	36.7 ± 1.67 ^b^	39.7 ± 1.76 ^b^	38.3 ± 0.33 ^b^	18.7 ± 1.20 ^b^	22.3 ± 2.91

^a–c^ Means denoted within the same column with different superscripts are significantly different at *p* < 0.05.

**Table 3 animals-10-00078-t003:** Effect of supplementing SHOTOR extender with vitamins, trace elements, and nanoparticles on viable, early apoptotic, apoptotic, and necrotic sperm in post-thawed epididymal camel spermatozoa using Annexin V/PI assay.

Treatment	Viable (%) (A−/PI−)	Early Apoptosis (%) (A+/PI−)	Late Apoptosis (%) (A+/PI+)	Necrosis (%) (A−/PI+)
Control	26.4 ± 1.10 ^g^	38.4 ± 0.26 ^a^	31.0 ± 1.30 ^a^	4.3 ± 0.07
Vitamin C	57.4 ± 0.92 ^c^	24.0 ± 0.55 ^c^	15.7 ± 1.30 ^c^	2.9 ± 0.18
Vitamin E	52.4 ± 0.58 ^d^	24.2 ± 0.61 ^c^	15.9 ± 0.18 ^c^	7.5 ± 0.20
SeNPs	78.1 ± 0.58 ^a^	12.0 ± 0.43 ^d^	4.3 ± 0.17 ^e^	5.6 ± 0.03
Na_2_SeO_3_	47.0 ± 0.09 ^e^	38.2 ± 0.32 ^a^	12.8 ± 0.30 ^d^	2.0 ± 0.12
ZnONPs	69.6 ± 0.26 ^b^	4.5 ± 0.35 ^e^	20.0 ± 0.35 ^b^	5.9 ± 0.26
ZnSO_4_	42.5 ± 0.64 ^f^	35.0 ± 0.23 ^b^	21.4 ± 0.43 ^b^	1.2 ± 0.03

^a–g^ Means denoted within the same column with different superscripts are significantly different at *p* < 0.05.

**Table 4 animals-10-00078-t004:** Effect of supplementing SHOTOR extender with vitamins, trace elements, and nanoparticles on the antioxidant (GSH, SOD) and oxidative biomarkers (malondialdehyde, MDA) in the extender of post-thawed epididymal camel spermatozoa.

Treatment	GSH (mg/dL)	SOD (U/mL)	MDA (nmol/mL)
Control	0.45 ± 0.03 ^e^	48.3 ± 5.93 ^f^	30.1 ± 2.15 ^a^
Vitamin C	0.59 ± 0.03 ^dc^	113.4 ± 5.66 ^dc^	14.8 ± 0.26 ^d^
Vitamin E	0.64 ± 0.02 ^bc^	124.3 ± 3.94 ^c^	14.8 ± 0.17 ^d^
SeNPs	0.77 ± 0.04 ^a^	168.0 ± 6.67 ^a^	12.7 ± 0.70 ^d^
Na_2_SeO_3_	0.55 ± 0.01 ^d^	106.8 ± 1.70 ^d^	18.4 ± 1.07 ^c^
ZnONPs	0.70 ± 0.00 ^ab^	148.2 ± 2.33 ^b^	13.6 ± 0.48 ^d^
ZnSO_4_	0.51 ± 0.03 ^de^	67.2 ± 6.28 ^e^	22.6 ± 1.32 ^b^

^a–f^ Means denoted within the same column with different superscripts are significantly different at *p* < 0.05.

**Table 5 animals-10-00078-t005:** Effects of supplementing SHOTOR extender with vitamins, trace elements, and nanoparticles on the sperm plasma membrane (PM) post-thawing.

Treatment	Intact PM	Slightly Swollen PM	Swollen PM	Lost PM
Control	38 ± 4.88 ^b^	12 ± 3.27 ^ab^	37 ± 4.85 ^a^	13 ± 3.38
Vitamin C	58 ± 4.96 ^ab^	7 ± 2.56 ^b^	23 ± 4.23 ^b^	12 ± 3.27
Vitamin E	55 ± 5.00 ^ab^	8 ± 2.73 ^b^	26 ± 4.41 ^ab^	11 ± 3.14
SeNPs	68 ± 4.69 ^a^	9 ± 2.88a ^b^	15 ± 3.59 ^b^	8 ± 2.73
Na_2_SeO_3_	53 ± 5.02 ^ab^	18 ± 3.86 ^a^	18 ± 3.86 ^b^	11 ± 3.14
ZnONPs	65 ± 4.79 ^ab^	7 ± 2.56 ^b^	17 ± 3.78 ^b^	11 ± 3.14
ZnSO_4_	51 ± 5.02 ^bc^	16 ± 3.68 ^ab^	21 ± 4.09 ^b^	12 ± 3.27

^a–c^ Means denoted within the same column with different superscripts are significantly different at *p* < 0.05.

**Table 6 animals-10-00078-t006:** Effects supplementing SHOTOR extender with vitamins, trace elements, and nanoparticles on the sperm acrosomes post-thawing.

Treatment	Intact Acrosome	Typical AR	Atypical AR	Lost Acrosome
Control	61 ± 4.90	24 ± 4.23	10 ± 2.88	5 ± 2.19
Vitamin C	68 ± 4.69	16 ± 3.68	12 ± 3.27	4 ± 1.97
Vitamin E	67 ± 4.73	18 ± 3.86	10 ± 3.02	5 ± 2.19
SeNPs	78 ± 4.16	10 ± 3.02	8 ± 2.73	4 ± 1.97
Na_2_SeO_3_	70 ± 4.61	16 ± 3.68	9 ± 2.88	5 ± 2.19
ZnONPs	74 ± 4.41	13 ± 3.38	10 ± 3.02	3 ± 1.71
ZnSO_4_	72 ± 4.51	12 ± 3.27	10 ± 3.02	6 ± 2.39

Intact acrosome: where sperm heads exhibited intact acrosomal membrane surrounding the acrosomal ground substance; Acrosome reaction (AR): a swelling of acrosomal ground substance with vesicles of fused plasma and outer acrosomal membranes; Atypical AR: sperm head presenting swelling of acrosomal ground substance dispersed under the swollen outer acrosomal.

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
