# Peer review of "Comparison between the Effects of Adding Vitamins, Trace Elements, and Nanoparticles to SHOTOR Extender on the Cryopreservation of Dromedary Camel Epididymal Spermatozoa"

_animals, 2020, doi:10.3390/ani10010078_

Round 1
Reviewer 1 Report
The manuscript by Shahin et al, entitled “Effects of supplementing SHOTOR extender with vitamins and nano-sized trace elements on the cryopreservation of dromedary camel epididymal spermatozoa" demonstrate the effects of different supplements (vitamins C and E, and nanoparticles zinc oxide and selenium) to the semen extender of camel epididymal spermatozoa during cooling and freezing/thawing cryopreservation. The authors demonstrated that supplementation of SHOTOR extender with zinc oxide and selenium nanoparticles lead to improved progressive motility, viability, and anti-oxidative defence, and reduced the ultrastructural abnormalities in camel epididymal spermatozoa. Since this study approach may be interesting, the paper deserves to be accepted but some minor revisions are necessary.
Many typewrite, and space mistakes are present throughout the manuscript (e.g. Line 88: please, correct the reference style) Figure 2: describe panel C Figure 3: what is the difference between normal nuclei and intact nuclei? Sperm showed in Figure 3 refer to what treatment? Sperm from control and from each treatment used in this study should be shown, in order to support the reduction in sperm abnormalities in treated specimens, as declared by the authors. In order to properly evaluate the integrity of nuclei, the authors should assess the DNA fragmentation with an appropriate test (ex by FACS).Author Response
The manuscript by Shahin et al, entitled “Effects of supplementing SHOTOR extender with vitamins and nano-sized trace elements on the cryopreservation of dromedary camel epididymal spermatozoa" demonstrate the effects of different supplements (vitamins C and E, and nanoparticles zinc oxide and selenium) to the semen extender of camel epididymal spermatozoa during cooling and freezing/thawing cryopreservation. The authors demonstrated that supplementation of SHOTOR extender with zinc oxide and selenium nanoparticles lead to improved progressive motility, viability, and anti-oxidative defence, and reduced the ultrastructural abnormalities in camel epididymal spermatozoa. Since this study approach may be interesting, the paper deserves to be accepted but some minor revisions are necessary.
Response:
Thank you very much for your careful revision. We find the comments are very helpful and positive for enriching the output of our manuscript. Below is our response to your comments after careful consideration and editing. We hope our revised version is satisfactory and suitable for publication in Animals.
Q1. Many typewrite, and space mistakes are present throughout the manuscript (e.g. Line 88: please, correct the reference style).
R1. We revised the whole manuscript and correct typewrite, and space mistakes.
Q2. Figure 2: describe panel C.
R2. We provided detailed description of panels C-H.
Q3. Figure 3: what is the difference between normal nuclei and intact nuclei? Sperm showed in Figure 3 refer to what treatment? Sperm from control and from each treatment used in this study should be shown, in order to support the reduction in sperm abnormalities in treated specimens, as declared by the authors.
R3. We modified the figure to show and clarify the parameters studied only in the plasma membrane and acrosome by TEM.
Q4. In order to properly evaluate the integrity of nuclei, the authors should assess the DNA fragmentation with an appropriate test (ex by FACS).
R4. We thank the reviewer for this suggestion, but we could not perform this analysis in the current work however, we will consider this essential protocol for our coming work.
Reviewer 2 Report
The submitted article animals-656023 intitled: “Effects of supplementing SHOTOR extender with vitamins and nano-sized trace elements on the cryopreservation of dromedary camel epididymal spermatozoa”, is great work that will give an important highlight on the applications of nanoparticle technology as supplementation in the used extender in order to improve the sperm cryotolerance and its quality after thawing especially for dromedary camel.
Indeed, this article is elucidating some innovative ideas using so much of analysis technics to evaluate the effect of each supplementation including vitamins (Vit E and Vit C), trace elements (Na2SeO3 (Sodium Selenite) and ZnSO4 (Zinc Sulfate)) and particularly the nanoparticles (SeNPs and ZnONPs).
However, that much of analysis evaluations are not all necessary and especially elucidated all in one article. Personally, I call the authors to see the possibility to divide the results into two articles defining part 1 and part 2 of “Effects of supplementing SHOTOR extender with Se or ZnO nanoparticles compared to vitamins and trace elements on cryotolerance of dromedary camel epididymal spermatozoa”.
In another hand, there are lot of results of this study that are so interesting to explain but unfortunately, they are not well presented, disorganized and superficially discussed at the end without giving the real value of this great condensate work.
Indeed, this version needs major modifications:
Title: It needs to be corrected since it makes the meaning ambiguous understanding at first place that there is just one supplementation to SHOTOR extender including vitamins and nano-sized trace elements at once and there is not the comparison with trace elements. Abstract: The same ambiguous meaning is still present in the abstract especially in line 18 then in line 20 “supplementation … with zinc oxide and selenium nanoparticles”, and in line 26. Line 26-28: Correct the sentence “the effects of supplementation … SeNPs)”. The classification of different supplementations used needs to be homogeneous in the total of the manuscript even in the results. For example (as used at the beginning of this article), at first place “Vitamins (Vit C and Vit E)”, then, “trace elements (Na2SeO3 (Sodium Selenite) and ZnSO4 (Zinc Sulfate))” and “nanoparticles (SeNPs (Selenium nanoparticles) and ZnONPs (Zinc Oxide nanoparticles))”. The same chosen notation must be present in the whole manuscript. Line 29: correct “livability” with vitality (as in the whole manuscript) Line 30: “at 5°C for 2h”. It is more interesting to show your findings post-thawing despite those at 5°C for 2h. Line 33: Why just GSH results are showed here while there are those of SOD too? Line 36: “SeNPs and/or ZnONPs”, suppress “and” because you did not yet show the supplementation feasibility of the both components especially that their zeta potential are opposite. Introduction: Line 67, 70 and 71: “Ascorbic acid”. You chose at first to use the appellation of Vitamin C and nor Ascorbic acid. If you choose to use this appellation, it would be preferable to use it in the whole manuscript and the same thing for Vitamin E with “Tocopherol alpha” appellation. Line 80: After Vitamins, it would be suitable to cite the bibliography of different studies that used supplementation of trace elements, before to talk about nanoparticles. Line 94: “viability”, correct it with vitality. Line 101-103: “Selenium … GPxs”. Are you talking about Selenium as trace element or as nanoparticle? If it is elucidated as trace element, it needs to place it before nanoparticles paragraph. Line 115: “Vitamin C…”. The trace elements (Na2SeO3 and ZnSO4) are missed in this sentence. Materials and Methods: The part of characteristics of nano-sized elements needs to be rewritten. Why the particle size difference is so big between ZnO and Se nanoparticles? Why the used concentration of ZnONPs is different from SeNPs? Why there is a need to make some voltage at 160KV? What is the used diluent? Generally, when nanoparticles are bigger than 10nm, their effect is hard to control. So, how can you explain the used particle size and for what reason? Line 149-152: why the used concentration of different supplementations is different and for what reason? Because, the effect could be just due to concentration and not to the component in itself. Use the same unit for all supplementations. Line 156: correct “livability” by “vitality”, and “abnormality” by “morphology or abnormal morphology”. Line 164: correct “motilities” by “motility rate”. Line 168: correct “sperm” by “spermatozoa”. Results The part of particle size, zeta potential and ultramorphology of the nanoparticles with Figure 1, must be placed in materials and methods in the part of characteristics of nano-sized elements. Table 1 can be removed while the results of this part of effect after cooling can be described briefly just in the text (line 242-248). For table 2, 3, 5 and 6, could be preferably represented in two histograms showing at first place the effect on sperm parameters including progressive motility rate, vitality rate (despite livability), morphology rate (it is rate of spermatozoa with normal morphology, despite using abnormality and cytoplasmic droplet). Then, at second place, the effect on functional sperm parameters including plasma membrane integrity rate, and apoptosis rate (despite to use all the results), intact plasma membrane rate and intact acrosome rate. Make the name of supplementations homogeneous in the whole manuscript with same classification. Table 4: homogenize the used units of GSH, SOD and MDA. Figure 2: Why there is use of SEM while the evaluation of spermatic morphologic can be realized just by an optical microscope? Figure 3: Select just the images of spermatozoa with intact acrosome versus with lost acrosome, and intact plasma membrane versus with abnormal one. Discussion Line 348: Compare the results based on the dose while yours was 1g/ml. Line 354: Compare the results based on the dose while yours was 50g/ml. The discussion needs much improvement since there is just simple elucidation of some studies without comparing their results with yours to reach some deep explanation of your findings with scientific reflection ReferencesWrite adequately some references in the text. They need correction.
Author Response
The submitted article animals-656023 entitled: “Effects of supplementing SHOTOR extender with vitamins and nano-sized trace elements on the cryopreservation of dromedary camel epididymal spermatozoa”, is great work that will give an important highlight on the applications of nanoparticle technology as supplementation in the used extender in order to improve the sperm cryotolerance and its quality after thawing especially for dromedary camel.
Indeed, this article is elucidating some innovative ideas using so much of analysis technics to evaluate the effect of each supplementation including vitamins (Vit E and Vit C), trace elements (Na2SeO3 (Sodium Selenite) and ZnSO4 (Zinc Sulfate)) and particularly the nanoparticles (SeNPs and ZnONPs).
Response:
We appreciate the time, efforts, and suggestions by the reviewer. All the suggestions have been addressed, and corrections made accordingly. We find the comments are very helpful and positive for enriching the output of our manuscript. Below is our response to your comments after careful consideration and editing. We hope our revised version is satisfactory and suitable for publication in Animals.
Q1. However, that much of analysis evaluations are not all necessary and especially elucidated all in one article. Personally, I call the authors to see the possibility to divide the results into two articles defining part 1 and part 2 of “Effects of supplementing SHOTOR extender with Se or ZnO nanoparticles compared to vitamins and trace elements on cryotolerance of dromedary camel epididymal spermatozoa”.
R1. We thank the reviewer for this suggestion; however, the division of the results will reduce the novelty of the current research because in our previous report showed the effects of vitamins on camel epididymal spermatozoa (https://scialert.net/abstract/?doi=ajas.2016.147.153). We aimed by the current paper to sum up and compare different formulations for camel sperms cryopreservation.
Q2. In another hand, there are lot of results of this study that are so interesting to explain but unfortunately, they are not well presented, disorganized and superficially discussed at the end without giving the real value of this great condensate work.
R2. We revised discussion section and did our best to represent, reorganize and explain the major results.
Indeed, this version needs major modifications:
Q3. Title: It needs to be corrected since it makes the meaning ambiguous understanding at first place that there is just one supplementation to SHOTOR extender including vitamins and nano-sized trace elements at once and there is not the comparison with trace elements.
R3. We thank the reviewer for this good suggestion. We edited the title accordingly.
Q4. Abstract: The same ambiguous meaning is still present in the abstract especially in line 18 then in line 20 “supplementation … with zinc oxide and selenium nanoparticles”, and in line 26. Line 26-28: Correct the sentence “the effects of supplementation … SeNPs)”.
R4. Corrected accordingly.
Q5. The classification of different supplementations used needs to be homogeneous in the total of the manuscript even in the results. For example (as used at the beginning of this article), at first place “Vitamins (Vit C and Vit E)”, then, “trace elements (Na2SeO3 (Sodium Selenite) and ZnSO4 (Zinc Sulfate))” and “nanoparticles (SeNPs (Selenium nanoparticles) and ZnONPs (Zinc Oxide nanoparticles))”. The same chosen notation must be present in the whole manuscript.
R5. We thank the reviewer for this good comment. We followed the suggestion and unified the nomenclature.
Q6. Line 29: correct “livability” with vitality (as in the whole manuscript)
R6. Corrected accordingly.
Q7. Line 30: “at 5°C for 2h”. It is more interesting to show your findings post-thawing despite those at 5°C for 2h.
R7. We thank the reviewer for this notion. However, we think that we must show the results of both before freezing and after cooling in the abstract to show that the benefit effect of treatments was noticed pre-freezing and post-thawing.
Q8. Line 33: Why just GSH results are showed here while there are those of SOD too?
R8. We apologize for this error, we added it accordingly.
Q9. Line 36: “SeNPs and/or ZnONPs”, suppress “and” because you did not yet show the supplementation feasibility of the both components especially that their zeta potential are opposite.
R9. We followed this suggestion and corrected it accordingly.
Q10. Introduction: Line 67, 70 and 71: “Ascorbic acid”. You chose at first to use the appellation of Vitamin C and nor Ascorbic acid. If you choose to use this appellation, it would be preferable to use it in the whole manuscript and the same thing for Vitamin E with “Tocopherol alpha” appellation.
R10. We unified the use of Vitamin C and E throughout the manuscript.
Q11. Line 80: After Vitamins, it would be suitable to cite the bibliography of different studies that used supplementation of trace elements, before to talk about nanoparticles.
R11. We thank the reviewer for the suggestion. We add two paragraphs about Zn and Se.
Q12. Line 94: “viability”, correct it with vitality.
R12. Corrected.
Q13. Line 101-103: “Selenium … GPxs”. Are you talking about Selenium as trace element or as nanoparticle? If it is elucidated as trace element, it needs to place it before nanoparticles paragraph.
R13. We clarified it in the revised manuscript.
Q14. Line 115: “Vitamin C…”. The trace elements (Na2SeO3 and ZnSO4) are missed in this sentence.
R14. We added them.
Q15. Materials and Methods: The part of characteristics of nano-sized elements needs to be rewritten. Why the particle size difference is so big between ZnO and Se nanoparticles? Why the used concentration of ZnONPs is different from SeNPs? Why there is a need to make some voltage at 160KV? What is the used diluent? Generally, when nanoparticles are bigger than 10nm, their effect is hard to control. So, how can you explain the used particle size and for what reason? Line 149-152: why the used concentration of different supplementations is different and for what reason? Because, the effect could be just due to concentration and not to the component in itself. Use the same unit for all supplementations.
R15. The Nano sized materials ranged from 1 to 100 nm and the Nano trace element used in this study within this range. Examination of nanoparticles requires a high magnification force, so the voltage was raised to 160 kV. The used diluent explained with details in line 161 to 164.
Furthermore, we used the different concentrations depending on the best results in the bibliography.
Q16. Line 156: correct “livability” by “vitality”, and “abnormality” by “morphology or abnormal morphology”. Line 164: correct “motilities” by “motility rate”. Line 168: correct “sperm” by “spermatozoa”.
R16. Corrected accordingly.
Q17. Results The part of particle size, zeta potential and ultramorphology of the nanoparticles with Figure 1, must be placed in materials and methods in the part of characteristics of nano-sized elements.
R17. We followed the suggestion, and moved the figure accordingly.
Q18. Table 1 can be removed while the results of this part of effect after cooling can be described briefly just in the text (line 242-248).
R18. In our view, we think it is better to leave it because studying the characteristics of sperm after the equilibrium period and before freezing is important to clarify whether the treatments have a harmful or benefit effect before freezing. If we noticed harmful effect, there is no need for freezing.
Q19. For table 2, 3, 5 and 6, could be preferably represented in two histograms showing at first place the effect on sperm parameters including progressive motility rate, vitality rate (despite livability), morphology rate (it is rate of spermatozoa with normal morphology, despite using abnormality and cytoplasmic droplet). Then, at second place, the effect on functional sperm parameters including plasma membrane integrity rate, and apoptosis rate (despite to use all the results), intact plasma membrane rate and intact acrosome rate.
R19. We think that using of histograms to show comparison between few groups is preferable. However, in the current study, we compared between 7 groups and we think that histograms in our case is not preferable. About cytoplasmic droplet, we think it is important because we use epididymal sperms. About details of plasma membrane integrity rate, and apoptosis rate, we think that it is best to present each test separately to clarify the results and clarify the degree of damage. For example: the necrosis is different from apoptosis. Therefore, if we present only viable sperm % we will loss valuable data. Therefore, we think that these details are helpful of researchers.
Q20. Make the name of supplementations homogeneous in the whole manuscript with same classification.
R20. Done
Q21. Table 4: homogenize the used units of GSH, SOD and MDA.
R21. These units were recommended according to the brochure used in the analysis by the manufacturer.
Q22. Figure 2: Why there is use of SEM while the evaluation of spermatic morphologic can be realized just by an optical microscope?
R22. That is true; however, SEM gives more accurate and detailed description.
Q23. Figure 3: Select just the images of spermatozoa with intact acrosome versus with lost acrosome, and intact plasma membrane versus with abnormal one.
R23. We followed the recommendation and replaced the figure and its legend accordingly.
Q24. Discussion Line 348: Compare the results based on the dose while yours was 1g/ml.
R24. We revised all discussion including this part.
Q25. Line 354: Compare the results based on the dose while yours was 50g/ml.
R25. We revised all discussion including this part.
Q26. The discussion needs much improvement since there is just simple elucidation of some studies without comparing their results with yours to reach some deep explanation of your findings with scientific reflection References
R26. We rewrote the discussion following your recommendation.
Q27. Write adequately some references in the text. They need correction.
R27. Done
Round 2
Reviewer 1 Report
The revision significantly improved the manuscript
Author Response
Response to Reviewer 1:
The revision significantly improved the manuscript
Response:
Thank you very much for your careful reviewing of the manuscript.
Reviewer 2 Report
I thank the authors for the improvement of this version of manuscript.
Otherwise, the discussion part still needs some improvement in interpretation of your results especially for nanoparticles of ZnO and Se.
In line 369 "higher surface reactivity", this issue needs to be developed showing why nanoparticles of trace elements are more effective than trace elements and vitamines.
After line 380, the importance of SeNPs needs to be elucidated after showing the effectiveness of ZnONPs.
In lines 381 and 382, you cited briefly how ZnONPs and SeNPs can protect the membrane integrity from peroxidation, while based on your results, there is some different between them. Try to compare their function and their effictiveness citing other studies that will help you to interpret deeply your results. This issue will highlight the utility of SeNPs or ZnONPs depending on the sperm quality.
Author Response
Response to Reviewer 2:
I thank the authors for the improvement of this version of manuscript. Otherwise, the discussion part still needs some improvement in interpretation of your results especially for nanoparticles of ZnO and Se.
Response:
We appreciate the time, efforts, and suggestions by the reviewer. All the suggestions have been addressed, and corrections made accordingly. We find the comments are very helpful and positive for enriching the output of our manuscript. Below is our response to your comments after careful consideration and editing. We hope our revised version is satisfactory and suitable for publication in Animals.
Q1. In line 369 "higher surface reactivity", this issue needs to be developed showing why nanoparticles of trace elements are more effective than trace elements and vitamines.
R1. We thank the reviewer for this notion. It has been edited and cited the appropriate reference.
Q2. After line 380, the importance of SeNPs needs to be elucidated after showing the effectiveness of ZnONPs.
R2. We thank the reviewer for the suggestion. It has been edited accordingly.
Q3. In lines 381 and 382, you cited briefly how ZnONPs and SeNPs can protect the membrane integrity from peroxidation, while based on your results, there is some different between them. Try to compare their function and their effictiveness citing other studies that will help you to interpret deeply your results. This issue will highlight the utility of SeNPs or ZnONPs depending on the sperm quality.
R3. We thank the reviewer for the suggestion. We showed the comparison and cited the appropriate references accordingly.